# Assessing the Performance of 18F-FDG PET/CT in Bladder Cancer: A Narrative Review of Current Evidence

**DOI:** 10.3390/cancers15112951

**Published:** 2023-05-28

**Authors:** Mara Bacchiani, Vincenzo Salamone, Eleana Massaro, Alessandro Sandulli, Riccardo Mariottini, Anna Cadenar, Fabrizio Di Maida, Benjamin Pradere, Laura S. Mertens, Mattia Longoni, Wojciech Krajewski, Francesco Del Giudice, David D’Andrea, Ekaterina Laukhtina, Shahrokh F. Shariat, Andrea Minervini, Marco Moschini, Andrea Mari

**Affiliations:** 1Oncologic Minimally Invasive Urology and Andrology Unit, Department of Experimental and Clinical Medicine, Careggi Hospital, University of Florence, 50121 Florence, Italy; 2Department of Urology, La Croix du Sud Hôpital, Quint Fonsegrives, 31000 Toulouse, France; 3Department of Urology, The Netherlands Cancer Institute, 1066 Amsterdam, The Netherlands; 4Department of Urology and Division of Experimental Oncology, URI, Urological Research Institute, IRCCS San Raffaele Scientific Institute, 20132 Milan, Italy; 5Department of Minimally Invasive and Robotic Urology, Wrocław Medical University, 50-367 Wroclaw, Poland; 6Department of Maternal-Infant and Urological Sciences, Policlinico Umberto I Hospital, Sapienza University of Rome, 00185 Rome, Italy; 7Comprehensive Cancer Center, Department of Urology, Medical University of Vienna, Vienna General Hospital, Währinger Gürtel 18-20, 1090 Vienna, Austria; 8Institute for Urology and Reproductive Health, Sechenov University, 119435 Moscow, Russia; 9Department of Urology, University of Texas Southwestern Medical Center, Dallas, TX 75390, USA; 10Department of Urology, Second Faculty of Medicine, Charles University, 15006 Prague, Czech Republic; 11Department of Urology, Weill Cornell Medical College, New York, NY 10065, USA; 12Karl Landsteiner Institute of Urology and Andrology, 1090 Vienna, Austria; 13Hourani Center for Applied Scientific Research, Al-Ahliyya Amman University, Amman 19328, Jordan

**Keywords:** bladder cancer, lymph node, FDG PET/CT, staging, restaging

## Abstract

**Simple Summary:**

Lymph node involvement is a prognostic determinant in the diagnostic work-up and management of muscle-invasive bladder cancer. Thus, it is crucial to provide an accurate staging of the bladder tumor to better identify the best therapeutic strategies to improve the chances of survival and the quality of life of patients affected by bladder cancer. Positron Emission Tomography/Computed Tomography (PET/CT) has been increasingly used in bladder cancer staging to improve the accuracy of lymph node detection and to overcome the lack of sensitivity and the understaging showed by conventional imaging. The aim of this narrative literature review is to provide an overview of the current evidence on the use of 18F-FDG PET/CT in the diagnosis, staging, and restaging of bladder cancer, with a particular focus on its sensitivity and specificity for the detection of LN metastasis. We aim to provide clinicians with a better understanding of 18F-FDG PET/CT’s potential benefits and limitations in clinical practice. Despite the heterogeneity of the studies in the literature and the lack of a consensus, 18F-FDG PET/CT provides important incremental staging and restaging information that can potentially influence the clinical management of patients affected by muscle-invasive bladder cancer.

**Abstract:**

Introduction: Lymph node (LN) involvement is a crucial determinant of prognosis for patients with bladder cancer, and an accurate staging is of utmost importance to better identify timely and appropriate therapeutic strategies. To improve the accuracy of LN detection, as an alternative to traditional methods such as CT or MRI, 18F-FDG PET/CT has been increasingly used. 18F-FDG PET/CT is also used in post-treatment restaging after neoadjuvant chemotherapy. The aim of this narrative literature review is to provide an overview of the current evidence on the use of 18F-FDG PET/CT in the diagnosis, staging, and restaging of bladder cancer, with a particular focus on its sensitivity and specificity for the detection of LN metastasis. We aim to provide clinicians with a better understanding of 18F-FDG PET/CT’s potential benefits and limitations in clinical practice. Materials and Methods: We designed a narrative review starting from a wide search in the PubMed/MEDLINE and Embase databases, selecting full-text English articles that have examined the sensibility and specificity of PET/CT for nodal staging or restaging after neoadjuvant therapy in patients with bladder cancer. The extracted data were analyzed and synthesized using a narrative synthesis approach. The results are presented in a tabular format, with a summary of the main findings of each study. Results: Twenty-three studies met the inclusion criteria: fourteen studies evaluated 18F-FDG PET/CT for nodal staging, six studies examined its accuracy for restaging after neoadjuvant therapy, and three studies evaluated both applications. To date, the use of F-18 FDG PET/TC for detection of LN metastasis in bladder cancer is controversial and uncertain: some studies showed low accuracy rates, but over the years other studies have reported evidence of high sensitivity and specificity. Conclusions: 18F-FDG PET/CT provides important incremental staging and restaging information that can potentially influence clinical management in MIBC patients. Standardization and development of a scoring system are necessary for its wider adoption. Well-designed randomized controlled trials in larger populations are necessary to provide consistent recommendations and consolidate the role of 18F-FDG PET/CT in the management of bladder cancer patients.

## 1. Introduction

Bladder Cancer (BCa) is a widespread disease that is ranked as the 10th most commonly diagnosed form of cancer globally. It has a notable effect on the wellbeing and longevity of individuals affected, in particular in situations where the cancer has progressed into a muscle invasive disease [1].

In fact, even if 70% of bladder cancer is represented by non-muscle-invasive tumors, the remaining 30% of patients have muscle-invasive bladder cancer (MIBC) associated with a high risk of lymph node (LN) involvement and distant metastases [2,3].

The presence of LN involvement in patients with BC, which indicates that cancer has progressed beyond the organ, is a crucial determinant of prognosis, with significant implications for treatment response and overall survival. The prognosis in cases of node-positive disease depends on multiple factors, including the stage and extent of the disease, as well as the presence of other risk factors such as advanced age, performance status, and comorbidities [4]. As such, a comprehensive evaluation of these factors is necessary to determine the optimal course of treatment and improve outcomes in affected individuals [5]. The management of node positive bladder cancer may involve a combination of surgery, chemotherapy, and/or radiation therapy.

It is important for these patients to receive an accurate staging of the bladder tumor to better identify timely and appropriate therapeutic strategies to improve their chances of survival and their quality of life [6]. Before a radical cystectomy, preoperative locoregional staging is important to define the indication to neoadjuvant chemotherapy (NAC) to eradicate any micro-metastatic disease or eventually to guide the clinician towards an extended pelvic LN dissection to improve the chances of performing a curative surgery. It is crucial to identify patients with advanced metastatic disease and carefully evaluate the indication for a salvage radical cystectomy [7]. If the surgery is not curative but only palliative, doctors and patients must consider the impact of a urinary diversion on the patient’s quality of life. Therefore, it is essential to weigh the potential benefits and drawbacks of the surgery and engage in a shared decision-making process to ensure that the patient’s goals and preferences are taken into account [8,9].

In routine clinical practice, preoperative staging of muscle-invasive bladder cancer (MIBC) typically involves performing a computed tomography (CT) scan of the chest, abdomen, and pelvis. Pelvic Magnetic Resonance Imaging (MRI) can also be used, although not routinely in clinical practice. However, despite their accuracy in detecting primary bladder disease, both CT and MRI have not proven to have high sensitivity for nodal staging. With a sensitivity of 50–85% for the detection of pelvic LN involvement, both CT and MRI understage about 1/3 of patients. CT and MRI remain valuable tools in the preoperative staging of muscle-invasive bladder cancer and can provide useful information for guiding treatment decisions. To improve the accuracy of LN detection, other imaging modalities such as positron emission tomography (PET) or sentinel LN biopsy may be considered in selected cases [10].

To overcome this lack of sensitivity in identifying lymph node involvement, Positron Emission Tomography/Computed Tomography (PET/CT) has been increasingly used in bladder cancer staging.

PET/CT is a non-invasive imaging modality which has gained increasing popularity in the evaluation of patients with cancer and provides whole-body imaging, even if imaging of chest, abdomen, and pelvis might be enough for many cancers [5].

PET/CT combines the functional information supplied by PET with the anatomic detail of CT, providing comprehensive information about the metabolic and structural changes in the body.

In particular, 18 F-fluoro-2-deoxy-D-glucose (18F-FDG) PET/CT confers information based on glucose uptake and identifies cells with a high uptake such as neoplastic cells with their increased utilization of glucose.

Since metabolic alterations occur before the morphological ones, 18F-FDG PET/CT enables an early detection of locoregional disease, distant metastases, and cancer recurrence, before they become evident by conventional imaging like TC or MRI.

Another important topic about MIBC is the post-treatment restaging after neoadjuvant chemotherapy.

In fact, an early evaluation of NAC response and of presence of residual disease is important to guide the perioperative management of patients. For example, patients with advanced disease and persistent LN involvement even after NAC have poor prognosis, and a multidisciplinary team has to evaluate whether to proceed with a radical cystectomy, intended for palliative rather than curative purpose. Patients who are unresponsive to neoadjuvant chemotherapy (NAC), particularly those with localized disease (cT2-T4aN0M0), might see more benefits from an immediate radical cystectomy rather than continuing NAC, especially considering chemotherapy-related side-effects.

Methods traditionally used for post-treatment restaging, such as cystoscopy, urine cytology, routine blood tests, CT, and MRI scans, do not have high diagnostic accuracy, and there is no consensus recommendation regarding restaging imaging during NAC.

18F-FDG PET/CT is already used to monitor response to NAC in other types of cancers, while its use in bladder cancer restaging is relatively new and controversial.

The aim of this narrative literature review is to provide an overview of the current evidence on the use of 18F-FDG PET/CT in the diagnosis, staging, and restaging of bladder cancer, with a particular focus on its sensitivity and specificity for the detection of LN metastasis. By synthesizing the available evidence on the use of 18-FDG PET/CT in MIBC, we aim to provide clinicians with a better understanding of its potential benefits and limitations in clinical practice.

## 2. Materials and Methods

### 2.1. Literature Search Strategy

A comprehensive literature search was performed in the PubMed/MEDLINE and Embase databases to identify relevant studies regarding the use of 18F-FDG PET/CT in the diagnosis, staging, and restaging of Muscle-Invasive Bladder Cancer (MIBC). Two independent authors (VS and MB) conducted the search using various combinations of the following terms: “PET/CT”, “18F-FDG PET/CT”, “bladder”, “locally advanced”, “cancer”, “tumour”, “carcinoma”, “pelvic lymph node”, and “staging”. The search strategy aimed to minimize bias and ensure a thorough review of the existing literature.

To further enhance the literature search, the reference lists of included studies and relevant reviews were manually screened to identify any additional pertinent publications. Additionally, a search for unpublished studies and conference abstracts was carried out to minimize the risk of publication bias.

### 2.2. Study Selection and Inclusion Criteria

Two authors (VS and MB) independently screened the titles and abstracts of the identified articles for relevance, and any disagreements were resolved by a third author (AM). Full-text articles were retrieved for those that met the following inclusion criteria:Studies that examined the sensitivity and specificity of PET/CT for nodal staging or restaging after neoadjuvant therapy in patients with bladder cancer.Studies published in English with no temporal restrictions.Studies conducted in humans.

Only English-language articles were considered eligible. Both prospective and retrospective clinical studies were included, such as cohort studies and case–control studies. Case reports, case series, and review articles were excluded from the review.

### 2.3. Quality Assessment of Included Studies

The methodological quality of the included studies was assessed independently by two reviewers (VS and MB) using appropriate quality assessment tools. For randomized controlled trials, the Cochrane Risk of Bias tool was employed, while the Newcastle–Ottawa Scale was used for cohort and case–control studies. Any discrepancies in quality assessment were resolved through discussion and consensus or, if necessary, by involving a third reviewer (AM).

### 2.4. Data Extraction and Analysis

The two reviewers independently extracted data from the included studies using a standardized form. The extracted data encompassed the following categories:Study design.Patient characteristics (age, gender, stage of disease, and histology).PET/CT parameters (radiotracer used, diuretic use, and acquisition protocol).Main findings of each study (sensitivity and specificity of PET/CT for nodal staging, Positive Predictive Value (PPV), Negative Predictive Value (NPV), and accuracy).

Any discrepancies in the extracted data were resolved through consensus. A narrative synthesis approach was employed for analyzing and synthesizing the extracted data. The results were presented in a tabular format, with a summary of the main findings of each study.

### 2.5. Synthesis of Results

The extracted data were synthesized using a narrative synthesis approach, taking into consideration the various factors that could influence the performance of 18F-FDG PET/CT in bladder cancer diagnosis, staging, and restaging. The results were presented in a tabular format, with a summary of the main findings of each study. Based on the findings of this narrative review and the identified gaps in the current literature, recommendations for future research were formulated.

## 3. Results

A total of twenty-three studies, published between 2005 and 2019, met the inclusion criteria for this literature review. Table 1 presents the characteristics of the included studies. Fourteen studies evaluated the sensitivity and specificity of 18F-FDG PET/CT for nodal staging, while six studies examined its accuracy for restaging after neoadjuvant therapy. Three studies evaluated both applications. All included studies reported the number of patients, with the range varying from a minimum of 15 to a maximum of 287 patients (Table 1). Gender distribution was reported in all studies except for one, with a male prevalence above 70% observed across the studies (Table 1).

Histology was reported in fifteen studies reported, of which eight studies presented only patients with Transitional Cell Carcinoma. Instead, seven studies included also patients with histological variants such as adenocarcinoma, squamous cell carcinoma, epidermoid, papillary, neuroendocrine, and paraganglioma (Table 1).

All included studies, except one, reported the timing of image uptake after the administration of 18F-FDG, with eighteen studies acquiring the first PET/CT images 60 min after administration. This standardized timing of image acquisition was assessed for ensuring consistency and accuracy in the interpretation of PET/CT results across studies. Two authors performed the first acquisition earlier, respectively, at 30 and 45 min; two authors posticipated the first acquisition, respectively, at 75 and 90 min.

Of the included studies, nine studies highlighted the use of furosemide as a diuretic, and eight studies specified the timing of PET/CT image acquisition following diuretic administration. In all the studies to evaluate the sensitivity and specificity of the technique, 18F-FDG PET/CT findings were compared to pathological reports; for distance metastases, moreover, it is necessary to also perform a comparison to follow-up imaging.

The distribution of studies reporting both staging classification by 18F-FDG PET/CT and pathological staging is presented in Table 2.

Anatomical pathology was the reference test used in all but one of the studies included. Clinical T stage was reported in seven (30%) of the twenty-three studies while clinical node and metastasis were reported in nineteen (82%) of the studies reviewed.

The pathological N stage was evaluated in twenty (86%) of the twenty-three studies included with an expected good and promising concordance between the clinical and the pathological data.

It should be noted that not all included studies reported both types of staging classification, highlighting the variability in reporting practices across studies. Sensitivity of 18F-FDG PET/CT was reported in all the studies, but not all authors provided interquartile range. All the studies provided specificity of 18F-FDG PET/CT, except one.

Sensitivity, specificity, PPV, NPV, and accuracy of the studies included are summarized in Table 3. Sensitivity ranged between 33% (Jensen et al.) and 100% in six studies (26%). Specificity ranged between 17% (Kollberg et al.) and 100% in five (21%) of the studies included.

The Positive Predictive Value (PPV) and Negative Predictive Value (NPV) were reported in seventeen (74%) of the twenty-three included studies, providing important information on the diagnostic accuracy of 18F-FDG PET/CT for bladder cancer. Additionally, thirteen (56%) of the twenty-three studies evaluated the accuracy of PET/CT through various parameters, such as different radiotracers, acquisition protocols, and patient characteristics. Sensitivity, specificity, PPV, NPV, and accuracy of the studies are summarized in Table 3.

## 4. Discussion

18F-FDG PET/CT has demonstrated promising results in detecting and staging various human cancers, as evidenced by previous studies [33]. However, the evidence regarding its use in bladder cancer is still a matter of debate. The use of 18F-FDG in primary bladder cancer detection is limited due to its high urinary excretion in the bladder and ureters which could be a confounding factor for detection of bladder wall lesions and metastatic regional LNS [34]. Identifying peri-vesical LN can be particularly challenging as they may be too small to be detected on CT or may be masked by adjacent 18F-FDG excretion in the urinary tract on PET/CT. The urinary excretion of 18F-FDG can lead to increased background activity in the pelvis, which can make it difficult to distinguish small LNs from surrounding urinary activity. As a result, accurate detection of peri-vesical LNs may require the use of more advanced imaging techniques, such as diffusion-weighted imaging (DWI) or dynamic contrast-enhanced MRI (DCE-MRI). Nonetheless, 18F-FDG PET/CT remains a valuable tool in identifying LN involvement and guiding treatment decisions in patients with bladder cancer.

Several measures may be employed to improve the accuracy of 18F-FDG PET/CT in bladder cancer imaging. These include oral pre-hydration to dilute the urinary tracer and the use of diuretics to enhance local regional accuracy in scans following furosemide administration. Bladder catheterization may also be effective in limiting 18F-FDG accumulation in the bladder and ureters, thereby improving the detection of bladder wall lesions and regional LNs. While these measures may improve the accuracy of 18F-FDG PET/CT in bladder cancer imaging, their routine use may not be practical or feasible in all patients, and their benefits should be weighed against potential risks and inconveniences [34,35]. To date, the use of F-18 FDG PET/TC for detection of LN metastasis in bladder cancer is controversial and uncertain: some studies showed low accuracy rates, but over the years other studies have reported evidence of high sensitivity and specificity [5,36].

Previous studies have suggested that the advantage of combined PET/CT over CT alone in detecting bladder cancer is minimal, likely due to the significant overlap between standardized uptake values (SUVs) of malignant lesions and active inflammatory processes. This overlap in SUVs can limit the specificity of 18F-FDG PET/CT in detecting bladder tumors and regional LNs, particularly in cases where there is significant inflammation or infection in the bladder or adjacent tissues.

While the combination of PET and CT imaging may still provide some additional information beyond what can be obtained with either modality alone, the limited advantage in diagnostic accuracy suggests that the routine use of combined PET/CT for bladder cancer imaging may not be justified in all cases.

A review of 2012 showed a sensitivity for combined PET-CT scan of 82% (95% CI: 0.72–0.89) [34].

Girard et al. in 2018 concluded that 18F-FDG PET/CT correctly detect LNs involvement in an additional 8% of patients compared to CT alone and that 18F-FDG PET/CT accuracy is 82% compared to 74% of CT alone [10]. Several studies have also shown that combined 18F-FDG PET/CT is superior for detection of distant metastases in bladder cancer [5,10]. The ability of PET to detect small metastases or LNs with high metabolic activity can increase the specificity of 18F-FDG PET/CT in detecting bladder cancer and regional metastasis [37].

According to Goodfellow et al., PET scans are considered useful if they result in a change in management for more than 10% of patients. In such cases, routine use of PET scans would be recommended. When PET scans lead to a change in management in 5–10% of patients, they should be used selectively for certain patients. However, if PET scans result in a change in management for less than 5% of patients, their routine use may not be justified. These recommendations highlight the need for careful consideration of the potential benefits and limitations of PET imaging, and the importance of individualized decision-making in the management of bladder cancer [22].

A recent review of 2022 on preoperative detection of pelvic LN involvement confirmed a higher sensitivity and specificity combining PET and CT scan when compared to the traditional imaging modalities [5]. A recent consensus statement by the European Association of Urology (EAU) and the European Society for Medical Oncology (ESMO) stated that 18F-FDG PET/CT should be included in oligometastatic disease staging to minimize the risk of overtreatment, when radical treatment options are being considered [37].

18F-FDG PET/CT has multiple applications beyond preclinical staging, including post-treatment restaging after neoadjuvant chemotherapy (NAC). Traditional modalities, such as cytology, cystoscopy, CT scans, MRIs, and routine blood tests, are often inaccurate in detecting residual disease or assessing treatment response. Early assessment of NAC response and/or residual disease is crucial for guiding perioperative management, limiting chemotherapy-related side effects and improving quality of life. 18F-FDG PET/CT is a reliable tool for monitoring response to chemotherapy in various cancer types and has been shown to be more accurate than conventional imaging. Recent studies have identified 18F-FDG PET/CT as an effective method for detecting both residual and recurrent disease, with superior accuracy in detecting post-treatment recurrence outside the urinary tract, primarily bone lesions, compared to conventional restaging techniques. Despite its potential benefits, there is currently no consensus on the use of 18F-FDG PET/CT for identifying chemo-sensitive bladder tumors during NAC. Nonetheless, the use of 18F-FDG PET/CT in post-treatment restaging may help to guide treatment decisions, such as whether to proceed with radical cystectomy or continue with NAC, based on the presence or absence of residual disease [5].

In addition to its potential role in bladder cancer staging, 18F-FDG PET/CT has applications in sentinel lymph node (SLN) mapping to aid in the resection of selected, invaded LNs during pelvic LND [38]. This approach can simplify histopathological examination and reduce the extent of LN dissection (LND) compared to blind template resection. Lymphoscintigraphy, CT, MRI, and fluoroscopy are commonly used methods for SLN mapping. SLN biopsy (SLNB) has been successfully used in breast and skin cancer treatment, contributing to a reduction in LND extent. Recent evidence suggests that 18F-FDG PET/CT combined with CT or MRI can be useful in evaluating LNs suspected to be involved based on CT/MRI findings, with a sensitivity of 92% and specificity of 91%. However, its sensitivity is considerably lower (7–23%) in patients with no suspicion of LN involvement on CT. Studies have shown that SLN mapping has a high detection rate and sensitivity in MIBC, particularly in patients with low pT stage bladder cancers and clinically negative LNs.

The complexity of bladder cancer staging using PET/CT prompts the need for a scoring system to improve its precision. This system should integrate key parameters like standardized uptake values (SUVs), indicative of tumor metabolic activity, and primary tumor size, which could correlate with disease progression. Lymph node involvement, particularly peri-vesical nodes, and the presence of distant metastases identified by PET/CT should also be incorporated due to their prognostic implications. Furthermore, changes in PET/CT findings post-neoadjuvant chemotherapy (NAC) could provide insights into tumor chemosensitivity and could thus influence the score. This score has been explained in detail in Figure 1.

The development of this scoring system calls for rigorous validation through future studies, correlating the score with patient outcomes in various settings and patient groups. Such a scoring system could reconcile PET/CT discrepancies and enhance its utility in bladder cancer assessment. However, ongoing research is essential to confirm the value of this proposed scoring system, as it presents a promising avenue to optimize PET/CT’s role in bladder cancer management.

Despite its potential benefits, 18F-FDG PET/CT has several limitations, including high cost, higher radiation exposure, lack of anatomic reference frame when performed alone, and prolonged lag time between PET and staging CT scans. A full-dose diagnostic staging CT with intravenous contrast medium may provide a better assessment of LNs and metastases than the CT component of the PET scan. False positive results can also lead to delay of treatment, unnecessary procedures, and additional costs. Therefore, PET scan results should be interpreted with caution in conjunction with CT scan and clinical judgment, particularly in cases of benign tumors or inflammatory lesions. Nonetheless, accurate staging information provided by 18F-FDG PET/CT can significantly influence therapeutic management and serve as an important prognostic indicator for progression-free survival (PFS) and overall survival (OS).

## 5. Conclusions and Future Directions

Although 18F-FDG PET/CT has shown promise as a feasible and reliable tool for bladder cancer staging, evidence for its use in diagnosis and staging is not yet strong. Like all radiological exams, 18F-FDG PET/CT is limited by the inability to retrieve a histological sample. Its non-invasive nature also makes it a potential tool for follow-up, but further studies are needed to evaluate its effectiveness in this setting. A standardized reporting system for characterizing bladder cancer using 18F-FDG PET/CT is still lacking, and efforts should be aimed at developing and validating a scoring system to improve its accuracy in staging. Despite these limitations, 18F-FDG PET/CT remains a valuable tool in the management of bladder cancer and can provide important information for treatment planning and prognostication.

In conclusion, 18F-FDG PET/CT provides important incremental staging information that can potentially influence clinical management in MIBC patients, although it can also lead to false positive results. However, supporting clinical evidence for its use is limited and requires further confirmation. Standardization of this approach and development of a scoring system are necessary for its wider adoption. In addition, 18F-FDG PET/CT has potential applications in sentinel LN mapping, which can guide the resection of selected and suspicious LNs during pelvic lymph node dissection. Sentinel LN biopsy is commonly used in breast and skin cancer treatment and can simplify histopathological examination and reduce the extent of LND.

## Figures and Tables

**Figure 1 cancers-15-02951-f001:**
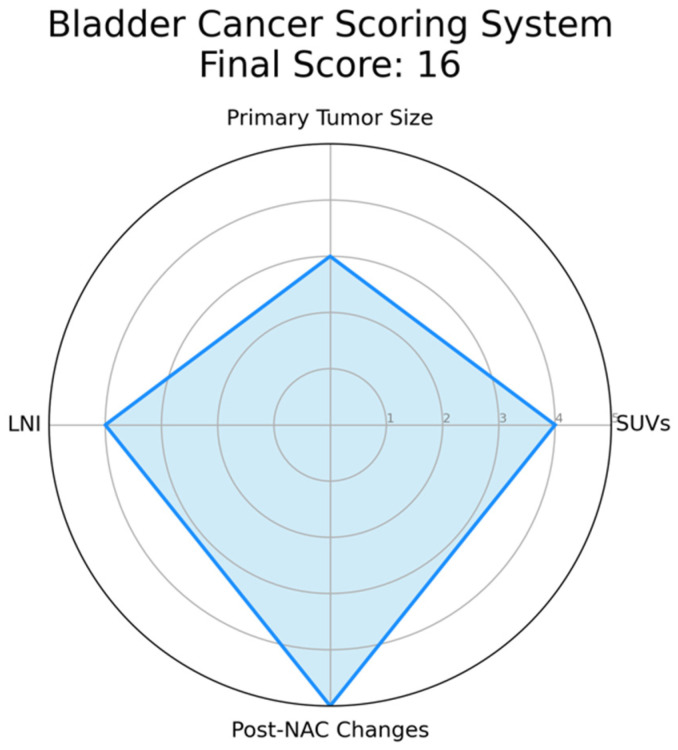
Example of a Radar Chart for the Proposed Bladder Cancer Scoring System. This chart represents an illustrative application of the proposed scoring system. Each axis corresponds to a parameter of the scoring system: standardized uptake values (SUVs), primary tumor size, lymph node involvement, and post-neoadjuvant chemotherapy (NAC) changes. The value along each axis represents the score for that parameter. SUVs is assigned as follows: 1 if SUV ≤ 2.5, indicating low metabolic activity and potentially less aggressive disease; 2 if 2.5 < SUV ≤ 5.0, suggesting moderate metabolic activity; 3 if 5.0 < SUV ≤ 7.5, suggesting relatively higher metabolic activity; 4 if 7.5 < SUV ≤ 10.0, indicative of high metabolic activity; 5 if SUV > 10.0, indicative of extremely high metabolic activity and potentially more aggressive disease. Primary tumor size is defined as follows: 1 if tumor size ≤ 2 cm, considered a small tumor; 2 if 2 cm < tumor size ≤ 4 cm, considered a moderately sized tumor; 3 if 4 cm < tumor size ≤ 6 cm, considered a relatively larger tumor, 4 if 6 cm < tumor size ≤ 8 cm, considered a large tumor, 5 if tumor size > 8 cm, considered a very large tumor, often associated with a worse prognosis. Lymph Node Involvement is defined as follows: 1 in case of no lymph nodes involved; 2 in case of single lymph node involvement; 3 in case of 2–3 lymph nodes involved, 4 in case of 4–6 lymph nodes involved, 5 in case of more than 6 lymph nodes involved, indicating extensive disease spread. Post-Neoadjuvant Chemotherapy (NAC) Changes are summarized as follows: 1 in case of no reduction in tumor size post-NAC, indicating no response; 2 in case of 1–25% reduction in tumor size post-NAC, suggesting a minimal response; 3 in case of 25–50% reduction in tumor size post-NAC, indicative of a partial response; 4 in case of 50–75% reduction in tumor size post-NAC, indicating a good response; 5 in case of more than 75% reduction or complete disappearance of the tumor post-NAC, indicating an excellent response. In this instance, hypothetical values are used for demonstration purposes: SUVs (4), Tumor Size (3), Lymph Node Involvement (4), and Post-NAC Changes (5). The filled area in the chart represents the composite score profile for a patient, providing a visual summary of the tumor’s characteristics according to the scoring system. Please note that in an actual case, values would be derived from clinical and imaging data, not randomly assigned.

**Table 1 cancers-15-02951-t001:** Characteristics of the included studies (publication year, patient selection, number of patients, gender distribution, histology, timing of image uptake, use of furosemide, and timing of image acquisition following diuretic administration).

Study	Year	Patient Selection (Detecting, Staging, Restaging)	N. Patients	Gender Male (%)	Histology	Image Time	Furosemide	Time after Furosemide Administration
Drieskens et al. [11]	2005	Staging	55	47 (85.5%)	TCC	60 min	20 mg	10 min
Anjos et al. [12]	2007	Detecting, staging Restaging	17	15 (88.2%)	TCC	60, 120 min	20 mg	60 min
Jadvar et al. [13]	2008	Restaging	35	25 (71.4%)	TCC	60 min	-	-
Swinnen et al. [14]	2009	Staging	51	43 (84.3%)	TCC	30 min	-	-
Kibel et al. [15]	2009	Staging, restaging	43	32 (74.4%)	TCC Adenocarcinoma Squamoso	60 min	20 mg	20 min
Lodde et al. [16]	2010	Staging	70	57 (81.4%)	TCC Epidermoide Neuroendocrino	75, 110 min	10 mg	30 min
Harkirat et al. [17]	2010	Staging, restaging	29	-	TCC	60, 150, 190 min	dosage not specified	60, 90 min
Apolo et al. [18]	2010	Detecting, restaging	57	38 (66.7%)	TCC, Adenocarcinoma Neuroendocrino Squamoso	60, 90 min	-	-
Jenses et al. [19]	2011	Staging	18	14 (77.8%)	TCC	60 min	-	-
Mertens et al. [20]	2013	Restaging	19	18 (94.7%)	-	60 min	-	-
Hitier-Berthault et al. [21]	2013	Staging	52	44 (84.6%)	TCC Adenocarcinoma Squamoso	60, 90 min	-	-
Goodfellow et al. [22]	2013	Staging	233	175 (75.1%)	TCC Adenocarcinoma Neuroendocrino Paraganglioma Squamoso	90 min	-	-
Nayak et al. [23]	2013	Staging	25	21 (84%)	-	45, 60 min	40 mg	120 min
Jeong et al. [24]	2015	Staging	61	46 75.4%	-	60 min	-	-
Aljabery et al. [25]	2015	Staging	54	47 (87%)	TCC	60, 90 min	-	-
Pichler et al. [26]	2016	Staging	70	53 (75.7%)	-	60 min	-	-
Uttam et al. [27]	2016	Staging	15	14 (93.3%)	-	60 min	20 mg	10–15 min
Soubra et al. [28]	2016	Staging	78	64 (81.1%)	-	60 min	40 mg	60 min
Alongi et al. [28]	2016	Restaging	41	36 (87.8%)	TCC Papillare Squamoso	60, 90 min	-	-
Kollberg et al. [29]	2017	Restaging	50	35 (78%)	-	-	20 mg	-
Zattoni et al. [30]	2017	Restaging	287	223 (77.7%)	TCC	60 min	-	-
Higashiyama et al. [31]	2018	Staging	25	19 (76%)	TCC Small cell carcinoma	60, 120 min	-	-
Girard et al. [10]	2019	Staging	61	56 (91.8%)	-	60 min	-	-

**Table 2 cancers-15-02951-t002:** Clinical and pathological staging. AP (Anatomical Pathology), FU (Follow-up).

Study	Year	Patient Selection (Detecting or Staging or Restaging)	Reference Test	Clinical t Stage	Clinical n Stage	Pathological t Stage	Pathological n Stage
Drieskens et al. [11]	2005	Staging	AP or FU	-	-	pT0: 0% pT1: 16% pT2: 47% pT3: 31% pT4: 6%.	-
Anjos et al. [12]	2007	Detecting Staging Restaging	APor FU	cT+: 35%	cN+: 47% cM+: 35%	pT+: 35%	pN0: 47% pM+: 35%
Jadvar et al. [13]	2008	Restaging	AP or FU	-	cN0: 34% cN+: 54%	-	pN0: 54% pN-: 34%
Swinnen et al. [14]	2009	Staging	AP	-	cN0: 86% cN+: 14%	pT0: 0% pT1: 24%; pT2: 43% pT3: 24%; pT4: 8%.	pN0: 75% pN+: 25%
Kibel et al. [15]	2009	Staging Restaging	AP	-	cN0: 79% cN+: 21%	-	pN0: 75% pN+: 25%
Lodde et al. [16]	2010	Staging	AP or FU	-	cN0: 82% cN1: 4% cN2: 14%	-	pN0: 3% pN1: 13% pN2: 12% pN3: 7%
Harkirat et al. [17]	2010	Staging Restaging	AP or FU	cT0: 45% cT+: 55%	cN+: 21% cM+: 7%	; -	pN+: 28% pM+: 14%
Apolo et al. [18]	2010	Detecting Restaging	AP or FU	cT0-T1: 23% pT2-3-4: 77%	-	pT0-T1: 12% pT2-T3: 44% pT4: 44%	-
Jenses et al. [19]	2011	Staging	AP	-	cN0: 99% cN+: 11%	-	pN0: 83% pN+: 17%
Mertens et al. [20]	2013	Restaging	AP	-	cN0: 63%	-	pN0: 74%
Hitier-Berthault et al. [21]	2013	Staging	AP	-	cN0: 77% cN+: 23%	pT0: 10% pTis: 4% pT1: 6% pT2: 13% pT3: 36% pT4: 31%	pN0: 58% pN+: 42%
Goodfellow et al. [22]	2013	Staging	AP or FU	-	cM0: 76% cM+: 24%	pTa: 7% pTis: 3% pT1: 26% pT2: 35% pT3: 21%; pT4: 8%	pN0: 88% pN+: 12%
Nayak et al. [23]	2013	Staging	AP	cT0: 4% cT+: 96%	cN0: 72%; cN+: 28%	pT0: 0% pT+:100%.	pN0: 64% pN+: 36%
Jeong et al. [24]	2015	Staging	AP	-	cN0: 69%; cN+: 31%	-	pN0: 72% pN+: 28%
Aljabery et al. [25]	2015	Staging	AP	cT0: 8% cT1:17% cT2: 19% cT3: 30% cT4: 26%	cN0: 88%; cN+: 22%	pT0: 0% pT1: 26% pT2: 18% pT3: 30% pT4: 26%	pN0: 69% pN+: 31%
Pichler et al. [32]	2016	Staging	AP	-	cN0: 83% cN+: 17%	pT0: 0% pTis: 3% pT1: 24% pT2: 35% pT3: 27% pT4: 11%	pN0: 84% pN1: 7% pN2: 7% pN3: 2%
Uttam et al. [26]	2016	Staging	AP	-	cN0: 47%; cN+: 53%	-	pN0: 80% pN+: 20%
Soubra et al. [27]	2016	Staging	AP (lynfh nodes and biopsy)	-	cM0: 90% cM+: 10%	pT0: 9% pTis: 14% pTa: 8% pT1: 8% pT2: 27% pT3: 19% pT4: 15%	pM0: 90% pM+: 10%
Alongi et al. [28]	2016	Restaging	AP or FU	cT1: 15% cT2: 10% cT3: 10% cT4: 10%	cN0: 49% cN+: 51%	pT0: 0% pT1: 29% pT2: 23% pT3: 25% pT4: 23%	pN0: 44% pN+: 56%
Kollberg et al. [29]	2017	Restaging	AP		cN0: 98% cN1: 2%		pN0: 86% pN1:14%
Zattoni et al. [30]	2017	Restaging	FU	-	-	pT0: 5% pTis: 6% pTa: 2% pT1: 9% pT2: 14% pT3: 34% pT4: 14% NA: 16%	pN0: 51% pN1: 12% pN2: 16% pN3: 4% pNx: 17 %
Higashiyama et al. [31]	2018	Staging	AP	cT0: 8% cT+: 92%	-	pT0: 0% pT+: 100%	-
Girard et al. [10]	2019	Staging	AP	-	cN0: 84%; cN+: 16%	pT0: 0% pT1: 25%; pT2: 16% pT3-4: 59%	pN0: 72% pN+: 28%

**Table 3 cancers-15-02951-t003:** Sensitivity, PPV (Positive Predictive Value), NPV (Negative Predictive Value), Accuracy, and Specificity.

Study	Year	Patient Selection (Detecting or Staging/Restaging)	Sensitivity (IQR)	PPV	NPV	Accuracy	Specificity (IQR)
Drieskens et al. [11]	2005	Staging	53% (27–79%)	75%	79%	78%	72% (0.51–0.88)
Anjos et al. [12]	2007	Detecting Staging Restaging	100% (54–100%)				100% (0.48–1.00)
Jadvar et al. [13]	2008	Restaging	100% (83–100%)				100% (0.78–1.00)
Swinnen et al. [14]	2009	Staging	46%	85%	84%	84%	97%
Kibel et al. [15]	2009	Staging Restaging	70% (35–93%)	78%	91%		94% (0.79–0.99)
Lodde et al. [16]	2010	Staging	57% (37–74%)	100%	80%	77%	100%
Harkirat et al. [17]	2010	Staging Restaging	87% (60–98%)	100%	78%		100% (0.59–1.00)
Apolo et al. [18]	2010	Detecting Restaging	81% (63–93%)				94% (0.70–1.00)
Jenses et al. [19]	2011	Staging	33%	50%	87%	93%	97%
Mertens et al. [20]	2013	Restaging	100%	94%	100%		67%
Hitier-Berthault et al. [21]	2013	Staging	36%	67%	65%	65%	87%
Goodfellow et al. [22]	2013	Staging	69%	87%	81%	86%	95%
Nayak et al. [23]	2013	Staging	100%				100%
Jeong et al. [24]	2015	Staging	47%	73%	82%		93%
Aljabery et al. [25]	2015	Staging	41%	58%	76%		86%
Pichler et al. [26]	2016	Staging	69%	50%	93%	84%	88%
Uttam et al. [27]	2016	Staging	100%	37%	100%		58%
Soubra et al. [28]	2016	Staging	56%			90%	98%
Alongi et al. [29]	2016	Restaging	87%	95%	85%	90%	94%
Kollberg et al. [30]	2017	Restaging	100%	88%	100%		17%
Zattoni et al. [31]	2017	Restaging	95%	95%	78%	91%	78%
Higashiyama et al. [32]	2018	Staging	92%				
Girard et al. [10]	2019	Staging	47%	80%	82%	82%	95%

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
