# Peer review of "Assessing the Performance of 18F-FDG PET/CT in Bladder Cancer: A Narrative Review of Current Evidence"

_cancers, 2023, doi:10.3390/cancers15112951_

Round 1
Reviewer 1 Report
The narrative review by Bacciani and Salamone et al discusses the accuracy of PET in staging muscle-invasive tumors in both the preoperative setting and post-neoadjuvant chemotherapy evaluation, which is a clinically relevant topic.
However, While useful, the review does not appear to bring anything new to the already published literature. Furthermore, the flow of the text loses clarity, making it difficult to draw effective clinical messages.
Paragraphing the results and discussion could help make the text clearer. Additionally, there are some areas that need clarification:
It seems to me that a few studies (recent and not) that might fit the inclusion criteria are missing.
(https://doi.org/10.1016/j.juro.2012.11.009, https://doi.org/10.1159/000528524, https://doi-org. /10.1080/21681805.2017.1321579).
Line 85: While I agree with the statement about CT scans, in my opinion, saying that MRI represents the routine clinical practice in BC staging does not properly represent clinical reality.
Line 89: The authors state "despite these limitations," but it is unclear which limitations they are referring to.
Line 115-117: These sentences need clarification. Please rephrase.
Line 127: Among the aims of the study, the authors state that they would have evaluated "various factors that may influence the accuracy of PET/CT, including disease stage, PET/CT parameters such as the type of radiotracer used, diuretic use, and acquisition protocol, as well as patient characteristics such as age and gender." The authors should explain how they intend to assess the influence of these parameters on the accuracy of PET/CT. Furthermore, the table does not include the age of the enrolled patients, which was one of the parameters the authors stated they wanted to evaluate. If such an evaluation is to be made, it would deserve a dedicated paragraph in the results section.
Was there a time limit for searching articles? When is the search up to date? Are there any more recent studies that fit the inclusion criteria?
Line 217: It would be better to have the number of studies reported instead of the percentage. Generally, it would be preferable to be consistent throughout the manuscript and always use either the absolute number, the percentage, or both.
Line 346: This last sentence can be removed since it is a repetition of what is already explained in the discussion.
Overall, the study provides valuable insights into the accuracy of PET in staging muscle-invasive tumors. However, the text needs some revisions and clarifications to improve the flow and enhance the clinical relevance of the findings.
Minor editing of English language required
Reviewer 2 Report
This is a very good review assessing the benefits of PET scans in bladder cancer for which agree there is still an ongoing debate therefore this is valuable contribution to the literature.
The manuscript though could be strengthen by a written description in the results summarizing table 2 and 3. For example, in the conclusion there is a discussion about the false positivity rate with the scans but the majority of the studies seem to indicate higher node positivity rate pathologically which is to the contrary. Are there discernible differences in studies which do show higher node positivity on scan? What is degree of the metastatic disease on found on PET scan in patient with clear node disease in these studies?
Also, the conclusion don't seem too full provide a consensus on the current status and seems to be a frequent back and forth on the good and the bad of the scans. For example, line 284(ref 35) it is stated that PET has a significant impact in 20% of patients without a further qualifying statement. This statement could be enough to stay there is clear benefit to the addition of the PET scan.
There really needs to be more discussion of the data with a clear recommendation going forward. For example, what would a potential scoring system look like (based on suv values, size of primary, etc.)
There are scattered minor grammatical errors. Overall, do think this would be a valuable contribution to the literature but could be significantly improved based on above comments.
There are scattered grammatical errors but quality is otherwise good.
Round 2
Reviewer 2 Report
Manuscript is very well written well very good response and corrections to prior review. No additional recommendations.